# Plant Coumarins with Anti-HIV Activity: Isolation and Mechanisms of Action

**DOI:** 10.3390/ijms24032839

**Published:** 2023-02-02

**Authors:** Ainur D. Sharapov, Ramil F. Fatykhov, Igor A. Khalymbadzha, Grigory V. Zyryanov, Oleg N. Chupakhin, Mikhail V. Tsurkan

**Affiliations:** 1Department of Organic and Biomolecular Chemistry, Ural Federal University, 620002 Yekaterinburg, Russia; 2Leibniz Institute of Polymer Research Dresden, 01005 Dresden, Germany

**Keywords:** coumarins, anti-HIV agents, reverse transcriptase inhibitors, integrase inhibitors, pyranocoumarins, furocoumarins, natural products

## Abstract

This review summarizes and systematizes the literature on the anti-HIV activity of plant coumarins with emphasis on isolation and the mechanism of their antiviral action. This review summarizes the information on the anti-HIV properties of simple coumarins as well as annulated furano- and pyranocoumarins and shows that coumarins of plant origin can act by several mechanisms: inhibition of HIV reverse transcriptase and integrase, inhibition of cellular factors that regulate HIV-1 replication, and transmission of viral particles from infected macrophages to healthy ones. It is important to note that some pyranocoumarins are able to act through several mechanisms or bind to several sites, which ensures the resistance of these compounds to HIV mutations. Here we review the last two decades of research on the anti-HIV activity of naturally occurring coumarins.

## 1. Introduction

Coumarin is a heterocyclic system of annulated pyrone and benzene rings. Coumarins are widely distributed in the plant kingdom, and are present in the roots, stems, bark, leaves, seeds, fruits, or flowers of various plant species. While unsubstituted coumarin can have significant hepatotoxicity [1], oxygenated coumarin derivatives generally have low toxicity [2] (expect aflatoxin coumarins), which may be due to the destructive opening of the pyrone ring [3] and hydroxylation [2] leading to the formation of easily excreted highly hydrophilic products. These properties offer great potential for combining coumarin derivatives with other compounds in the treatment of diseases [4,5]. This is especially important for the highly effective antiretroviral therapy proposed by David Ho in 1996 [6,7].

In addition, the synthesis of hydroxyl-containing coumarins by Pechmann condensation [8,9] is a convenient protocol, which makes this class of heterocyclic compounds particularly easy to access, both in terms of obtaining libraries of compounds to optimize the structure of the drug lead and in terms of large-scale industrial production.

Over the past 30 years, natural coumarin compounds have attracted great attention as lead compounds for developing low molecular weight antiviral agents, particularly anti-HIV agents [10,11]. Despite the fact that there are no approved coumarin derivatives in clinical practice, several natural and semi-synthetic coumarins have recently been discovered and are currently undergoing various phases of clinical trials [12].

The HIV replication cycle consists of thirteen steps (Figure 1) [13]; each step can be the target of chemotherapeutic interventions. The viral cycle begins with virion attachment to the membrane receptor of the T cell (step 1), followed by entry of viral components into the cell (step 2). Uncoated viral reverse transcriptase (RT) and viral RNA trigger reverse transcription (step 4), producing viral DNA, which is transported to the nucleus (step 5) and incorporated into the DNA of the infected cell (step 6). Viral RNA is synthesized in the nucleus (step 7), and the viral RNA leaves the nucleus (step 8) and serves to synthesize viral proteins (step 9). Then, the synthesized viral components assemble in new virions (step 10) and leave the infected cell (steps 11 and 12). After protease-catalyzed maturation (step 13), they may infect new cells.

Currently, the vast majority of antiretroviral drugs target the steps of reverse transcription and integration (steps 4 and 6). In 2022, there were 14 FDA-approved reverse transcriptase and integrase inhibitors, while only 7 drugs that aimed to block the remaining 11 stages had been approved [14]. RT inhibitors can be nucleoside or non-nucleoside RT inhibitors (NRTIs and NNRTIs). NRTIs contain a nucleoside base, which is incorporated into the growing DNA chain, terminating its elongation, while NNRTIs block the active site of the RT itself.

In terms of the mechanism of antiviral action, anti-HIV coumarins can be divided into four groups (Figure 2): NNRTIs (this group includes simple coumarins **I**, linear furocoumarins **IIa**, as well as both angular and linear pyranocoumarins **IIIa** and **IIIb**, respectively), the largest group; integrase inhibitors (coumestans **IIb**); compounds inhibiting cellular factors that regulate HIV-1 replication, for example, by inhibition of the transcription factor NF-κB (simple coumarins **I**, pyranocoumarins **IIIc**); and inhibitors of the transmission of viral particles from infected macrophages to healthy ones, for example, inhibiting HIV-1 entry, and downregulating the expression of chemokine receptors CXCR4, CD4, and CCR5 (simple coumarins **I**, furocoumarins **IIa**, and pyranocoumarins **IIIa**). In addition, for some compounds there are still no reliable data on the mechanism of action [15]. Therefore, typical coumarins with anti-HIV activity are RT and integrase inhibitors, so they can potentially be used as components of highly effective antiretroviral therapy.

Naturally occurring anti-HIV coumarins may be classified by their structure into three main groups: simple coumarins **I**, furocoumarins (psoralene-type **IIa** and coumestan-type **IIb**), and pyranocoumarins (angular **IIIa** and linear **IIIb**) (Figure 2) [16].

This review provides a full account of simple coumarins **I** and their annulated analogs **II** and **III** as potential anti-HIV agents, with emphasis on recent articles published in the last 5 years. Since not all coumarins have a known mechanism of anti-HIV action but all have a known structure, in this review, we have systematized the antiviral coumarins according to their structure.

## 2. Simple Coumarins

In the group of simple coumarins, we included compounds that contain the benzopyrone system and do not contain other cycles annulated with it; substituted coumarins are also included in this section. Simple natural coumarins usually contain methoxy, hydroxy groups and/or 1–3 prenyl residues as substituents of benzo or pyran rings via an optional -O-linker.

Methoxy- and hydroxy-derivatives of coumarin represent a small group of anti-HIV coumarins with moderate antiviral activity. 5-, 6-, 7-, and 8-methoxy- as well as 6- and 7-hydroxycoumarins are known. 

It was reported that aesculetin **1** (Figure 3) isolated from *Fraxinus ornus* (manna ash) [17], *Manihot esculenta* (cassava) [18] and *Artemisia capillaries* [19] was able to inhibit HIV replication in H9 cells with EC_50_ = 2.51 μM and a therapeutic index (TI) of 11.2 (cytotoxic concentration (CC_50_) = 28.1 μM) [19].

Citropten (5,7-dimethoxycoumarin) **2** and 6,8-dimethoxycoumarin **3** (Figure 3), found in the plant families *Campanulaceae* [20] and *Rutaceae* [21,22], showed moderate anti-HIV activity with EC_50_ = 33.8 μM and CC_50_ = 417 μM (TI = 12.4) and EC_50_ = 100.9 μM and CC_50_ > 970.9 μM (TI > 7.36), respectively [22], while 5,7,8-trimethoxycoumarin **4** (Figure 3) from *Zapota ailanthoides* showed high activity with EC_50_ = 0.933 μM and TI = 107 [23].

Various derivatives of terpene coumarins and related compounds isolated from plants of the families *Rutaceae* [22,23,24,25,26,27,28], *Umbelliferae* [29,30], *Fabaceae* [31,32] were found to be active against HIV (Table 1) [22,23,24,32]. Among them, auraptene **5** and anisocoumarin B **6** inhibit the replication of HIV-1 (strain LAV-1) with EC**_50_** activities of 5.7 µM and 23.2 µM, respectively, on peripheral blood mononuclear cells [24]. Regarding the HIV-1 IIIB strain, the activity of auraptene **5** was 10 times lower (EC_50_ = 59.7 μM) in the C8166 cell line [24]. Anisocoumarin B **6** inhibits HIV viral replication with EC_50_ 18.3 μM, with a sufficiently high TI > 44.4 [22]. In Sup T1 cells, Isobtusitin **7** had a weak HIV inhibitory effect (about 10%) with a concentration close to CC_50_, namely 20 μg/mL [33].

Farnesiferol C **8** (Table 1) is a well-known sesquiterpene coumarin isolated from genus *Ferula* (*Apiaceae*) species such as *Ferula assafoetida* and *Ferula szowitsiana*. It has been reported to have cytotoxic, apoptotic, MDR reversal, antitumor, antimutagenic, and antiviral activity [34]. Recently, farnesiferol C **8** was identified as an HIV-1 RT inhibitor with a mixed inhibition mechanism (IC_50_ = 30 μM). Using theoretical and experimental methods, it was demonstrated that farnesiferol C **8** could quench the intrinsic fluorescence emission of HIV-1 RT through a static quenching mechanism, while molecular docking studies indicated that farnesiferol C **8** interacts with the enzyme through a hydrophobic pocket [35]. 

**Table 1 ijms-24-02839-t001:** O-terpenoid anti-HIV coumarins.

Compound	Plant Source	Anti-HIV Activity and Toxicity
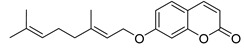 auraptene **5**	*Triphasia trifolia* [24]*Zanthoxylum schinifolium* [25]	EC_50_ = 5.7 μM CC_50_ = 46.8 μMTI = 8.21 [24]
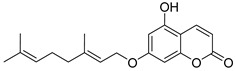 anisocoumarin B **6**	*Poncirus trifoliata* [22]*Clausena anisata* [31]	EC_50_ = 18.3 μM CC_50_ = 813 μMTI = 44.4 [24]
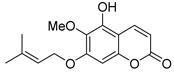 isobtusitin **7**	*Psiadia dentata* [33]	EC_10_ = 20 μM CC_50_ = 20 μM [33]
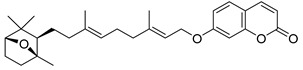 farnesiferol C **8**	*Ferula assa-foetida* [35]*Ferula szowitsiana* [36]*Ferula kopetdaghensis* [37]	EC_50_ = 30 μM [35]CC_50_ > 150 μM [34]

The low toxicity of coumarin O-terpenoids (up to 813 μM) allows for an acceptable TI in spite of the low EC_50_ (5 to 60 μM).

C-terpenoid anti-HIV coumarin derivatives are represented by a large number of compounds. In general, C-terpenoids are characterized by higher activity compared to their O- counterparts. Among them, it is possible to distinguish monoterpenoids and diterpenoids (Table 2).

3-(1,1-Dimethylallyl)-8-hydroxy-7-methoxycoumarin **9** (Table 2), the only representative of the C3-monoterpenoids, inhibits HIV viral replication with EC_50_ 10.7 μM, with a sufficiently high TI (31.0) [22]. 

C6-coumarin monoterpenoids include fipsomin **10** [38], phellodenol C **11** [39], isophellodenol C **12** [40], and coumarins **13**-**16** [41,42,43,44,45,46,47,48] (Table 2). C6-terpenoids are characterized by moderate anti-HIV activity (3–8 μM).

Osthol **17** and O-methylcedrelopsin **18**, C8-coumarin monoterpenoids, exhibited high inhibitory activity against HIV replication in H9 cells with EC_50_ values of 0.155 μg/mL (0.64 μM) and 0.576 μg/mL (2.10 μM), respectively, and TIs of 75.5 [30] and 36.6 [23], respectively.

Diterpene derivatives represent a large class of terpene coumarins, which usually have the highest anti-HIV activity. 

**Table 2 ijms-24-02839-t002:** C-terpenoid anti-HIV coumarins.

Compound	Plant Source	Anti-HIV Activity and Toxicity
Monoterpenoids		
C3-monoterpenoids **9**		
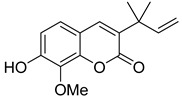 **9**	*Poncirus trifoliata* [22]	EC_50_ = 10.7 μM CC_50_ = 330 μM TI = 31 [22]
C6-monoterpenoids **10–16**		
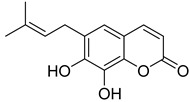 fipsomin **10**	*Ficus nipponica* [38] *Artocarpus heterophyllus* [45]	EC_50_ = 2.93 μM CC_50_ > 200 μM TI > 68.26 [45]
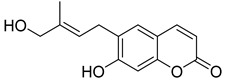 phellodenol C **11**	*Phellodendron amurense* [39] *Artocarpus heterophyllus* [45]	EC_50_ = 8.07 μM CC_50_ > 200 μM TI > 21.93 [45]
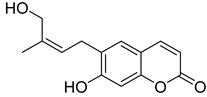 isophellodenol C **12**	*Heracleum candicans* [40] *Artocarpus heterophyllus* [45]	EC_50_ = 9.12 μM CC_50_ > 200 μM TI > 24.78 [45]
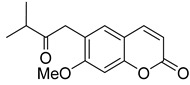 **13**	*Manilkara zapota* [41] *Ruta pinnata* [42] *Skimmia laureola* [43] *Acalypha indica* [44]	EC_50_ = 8.69 μM CC_50_ > 200 μM TI > 38.02 [41]
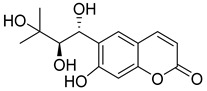 **14**	*Artocarpus heterophyllus* [45] *Brombya* sp. Nova [46]	EC_50_ = 5.68 μM CC_50_ > 200 μM TI > 35.21 [45]
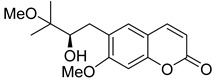 (+)-trachypleuranin A **15**	*Manilkara zapota* [41] *Harbouria trachypleura* [47]	EC_50_ = 5.26 μM CC_50_ > 200 μM TI > 38.02 [41]
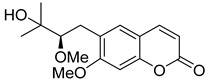 **16**	*Manilkara zapota* [41] *Skimmia laureola* [43] *Clausena lenis* [48]	EC_50_ = 6.73 μM CC_50_ > 200 μM TI > 29.72 [41] EC_50_ = 7.39 μM CC_50_ > 200 μM TI > 31.9 [48]
C8-monoterpenoids **17, 18**		
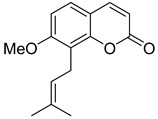 osthol **17**	Many species. For recent reviews, see [49,50]	EC_50_ = 0.64 μM CC_50_ = 48 μM TI = 75.5 [30]
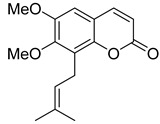 O-methylcedrelopsin **18**	*Zanthoxylum ailanthoides* [23]	EC_50_ = 7.66 μM CC_50_ = 77 μM TI = 36.6 [23]
Diterpenoids		
C3,C6-diterpenoids **19**–**27**		
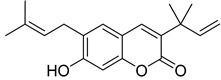 gravelliferone **19**	*Manilkara zapota* [41] *Clausena lansium* [51] *Helietta apiculata* [52]	EC_50_ = 2.28 μM CC_50_ > 200 μM TI > 87.7 [41]
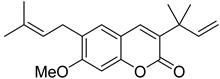 O-methyl gravelliferone **20**	*Manilkara zapota* [41] *Ruta pinnata* [53]	EC_50_ = 3.49 μM CC_50_ > 200 μM TI > 35.3 [41]
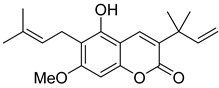 tanizin **21**	*Manilkara zapota* [41] *Artocarpus heterophyllus* [45] *Helietta apiculata* [52]	EC_50_ = 4.26 μM CC_50_ > 200 μM TI > 47.0 [41]EC_50_ = 0.56 μM CC_50_ > 200 μM TI > 357 [45]
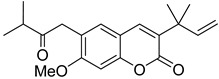 manizapotin A **22**	*Manilkara zapota* [41]	EC_50_ = 0.12 μM CC_50_ > 200 μM TI > 1667 [41]
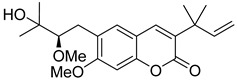 manizapotin B **23**	*Manilkara zapota* [41]	EC_50_ = 0.33 μM CC_50_ > 200 μM TI > 606 [41]
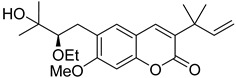 manizapotin C **24**	*Manilkara zapota* [41]	EC_50_ = 0.42 μM CC_50_ > 200 μM TI > 467 [41]
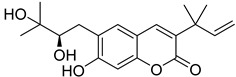 clauselenin B **25**	*Clausena lenis* [48]	EC_50_ = 0.68 μM CC_50_ > 200 μM TI > 294 [48]
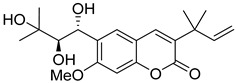 artoheteronin **26**	*Artocarpus heterophyllus* [45]	EC_50_ = 0.18 μM CC_50_ > 200 μM TI > 1111 [45]
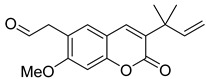 anisocoumarin A **27**	*Manilkara zapota* [41] *Artocarpus heterophyllus* [45]	EC_50_ = 0.97 μM CC_50_ > 200 μMTI > 206 [41] EC_50_ = 0.33 μM CC_50_ > 200 μMTI > 606 [45]
C6,C8-diterpenoids **28**		
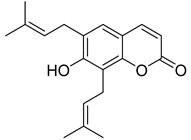 6,8-diprenylumbelliferone **28**	*Clausena lenis* [48]	EC_50_ = 1.59 μM CC_50_ > 200 μM TI > 125 [48]

C3,C6-diterpenoids are characterized by high activity against HIV-1 RT, with IC_50_ values 4-0.12 μM, in combination with low toxicity (CC_50_ > 200 μM) [32,41,45] (Table 2). Coumarin diterpenoids have been isolated from a large variety of plant materials [32,41,45,51,52,53].

Ethanol extract of the fruits of *Manilkara zapota* contains at least 7 coumarin diterpenoids with high anti-HIV activity: tanizin **21** (EC_50_ = 0.56 μM), manizapotins A, B, and C **24**, **25**, and **26** (EC_50_ = 0.12, 0.33, and 0.42 μM, respectively), anisocoumarin A **27** (EC_50_ = 0.33 μM), gravelliferone **19** and its methyl ether **20** (EC_50_ = 2.28 and 3.49 μM), as well as monoterpenoid coumarins **15** and **16** (EC_50_ = 5.26 and 6.37 μM, respectively). Gravelliferone **19,** O-methyl gravelliferone **20,** and tanizin **21** were also isolated from *Clausena lansium* [51], *Helietta apiculata* [52], *Ruta pinnata* [53], *Artocarpus heterophyllus* [45] and *Helietta apiculata* [52] species.

Clauselenin B **25**, together with diprenylated coumarin **28** isolated from stems of *Clausena lenis*, demonstrated high activity against HIV-1 RT (EC_50_ = 0.68 and 1.59 μM for compounds **25** and **28**, respectively) [32].

Another diprenylated coumarin, artoheteronin **26**, was isolated from fruits of *Artocarpus heterophyllus*. This compound demonstrated anti-HIV RT activity with EC_50_ = 0.18 μM [45]. 

Anisocoumarin E **29**, a representative of the C,O-diterpene coumarins (Figure 4), was described in 2010 as an HIV RT inhibitor with IC_50_ = 23.2 μM and CC_50_ = 96.8 μM (TI = 4.17) [24].

T. Okuda and colleagues demonstrated that 3-arylcoumarin, namely glycycoumarin **30** (Figure 4), isolated from licorice inhibits giant cell formation in HIV-infected cultures at 20 μg/mL without any observed cytotoxicity [32]. Later, J. Alcami et al. studied natural 4-phenyl-coumarins isolated from the *Marila pluricostata* plant (family *Clusiaceae*) for their activity against HIV transcription, which occurred by the mechanism of inhibition of the transcription factor NF-κB and by the function of transactivation of transcription (Tat). Among the studied coumarins, only two derivatives (disparinol A **31** and isodispar B **32** (Figure 4)) showed high inhibitory efficacy towards NF-κB proteins (81.0% and 55.4% at 25 μM respectively) and Tat (similar for both, ~40% at 25 μM and ~70% at 50 μM) [54].

In 2022, Fobofou et al. reported a new type of coumarin dimer with activity against HIV [15]. In particular, 8,8’-dicoumarin derivative (bichromonol) **33** (Figure 5) isolated from the stem bark of *Hypericum roeperianum* exhibits anti-HIV activity in infected MT-4 cells. Bichromonol demonstrated micromolar activity with EC_50_ = 6.6–12.0 µM against both wild-type HIV-1 and its clinically relevant mutant strains. Bichromonol is particularly more effective than the commercial drug nevirapine against the resistant variants A17 and EFVR, and might thus have the potential to serve as a new anti-HIV lead. Unfortunately, there are still no reliable data on the mechanism of anti-HIV action of bichromonol. Since simple coumarins typically inhibit HIV RT, it was suggested that bichromonol is an HIV RT inhibitor. However, preliminarily enzymatic assay reveals no RT inhibitory activity of compound **33**. 

For the anti-HIV activity of simple coumarins, the key factor is the presence of certain substituents. Since all simple coumarins inhibit HIV RT, a structure–activity relationship seems possible for them. Coumarins with only hydroxy or methoxy groups (compounds **1**–**3**) tend to have low anti-HIV activity; O-terpenoid and C-monoterpenoid coumarins **5**–**8** and **9**–**18** often have acceptable characteristics, while diterpenes **19**–**28** mostly have high activity (EC_50_ reaches the nanomolar range). The most active diterpene coumarins contain isoprene substituents at the C3 and C6 positions.

In addition, simple diterpene coumarins have a low molecular weight, and their activity is tolerant to the introduction of polar substituents (compounds **22**–**27**). Therefore, in our opinion, diterpene coumarins are promising lead compounds for developing new anti-HIV agents. In development, attention should probably be paid to unsaturated double bonds, which pose a potential problem in terms of drug stability and drug metabolism.

## 3. Furocoumarins

Furocoumarins are a class of compounds of natural and synthetic origin with various biological activities. Among natural furocoumarins, two classes of anti-HIV agents are distinguished: psoralene-type **IIa** and coumestan-type **IIb** (Figure 6). Due to the plane structure of psoralene, it intercalates into the DNA helix, forming strong bonds with nitrogenous bases [55]. This property is used in the photochemical treatment of skin diseases [55,56,57].

It was demonstrated that furocoumarins **34**–**43** inhibit HIV replication in H9 cells with an EC_50_ in the range of up to 0.37 μM (Table 3) [30,58]. Among them, the most active are psoralen **33** (EC_50_ = 0.54 and TI = 191), bergapten **35** (EC_50_ = 1.63 and TI = 70), imperatorin **36** (EC_50_ < 0.37 and TI > 1000) and heraclenol **40** (EC_50_ = 0.38 and TI = 870) [30,58]. Subsequently, Sancho and colleagues showed that imperatorin **36** showed no activity against RT and integrase [59]. However, it inhibits the critical transcription factor Sp1 and blocks infected cells in the G1 phase of the cell cycle. This indicates its potential therapeutic role in the treatment of HIV infections by inhibiting cellular factors that regulate HIV-1 replication at the transcriptional level [59].

Very recently, it was demonstrated that prenylated coumarins clauselenins A and C **44** and **45** (Figure 7) isolated from stems of *Clausena lenis* inhibit HIV RT with a half-effective concentration (EC_50_) of 290 and 170 nM, respectively. These coumarins **44** and **45** also showed good half-maximal inhibition (IC_50_) values of 0.52 and 1.06 μM. In addition, clauselenins A and C were non-toxic to normal C8166 cells (CC_50_ > 200 μM), and TI_50_ was > 689 [48]. Therefore, these compounds can serve as new leads in the development of new antiviral agents.

The second group of natural furocoumarins is the coumestans. They are mainly distributed in plants of the *Fabaceae* family [26,66,67,68,69] and have a wide spectrum of biological activity [70]. However, the antiviral activity of coumestans is poorly studied.

Wedelolactone **46** (Figure 8) isolated from *Eclipta prostrate* plants also demonstrated selective inhibition of HIV-1 integrase with IC_50_ = 4 μM [71]. It should be noted that wedelolactone is the only coumarin that has activity against HIV integrase. In addition, wedelolactone has weak protease inhibition activity (32.7% at 100 μM) [71].

Thus, among furocoumarins, linear coumarins are the largest class of anti-HIV compounds. The antiviral action is mediated by host cell cycle inhibition. The only example of integrase inhibitors is wedelolactone.

## 4. Pyranocoumarins

Pyranocoumarins are an important subclass of coumarins with a wide spectrum of biological activity, among which anti-HIV activity can be distinguished [72,73,74].

In this section, linear and angular pyranocoumarins are discussed.

Linear pyranocoumarins **47**–**49** (Figure 9) isolated from the stems of *Clausena lenis* [48] and *Ficus nervosa* [75] demonstrated high to moderate anti-HIV activity in the micromolar range (EC_50_ = 1.87–3.19 μM) and TI = 63–107 [54]. As demonstrated by the enzyme assay, the anti-HIV activity of these pyranocoumarins is mediated by RT inhibition [48].

In addition to anti-HIV activity, these pyranocoumarins demonstrate antimycobacterial [75] and anti-inflammatory activity [48].

Angular pyranocoumarins comprise tetracyclic calanolide and tricyclic seseline-type scaffolds.

Among the natural pyranocoumarins, tetracyclic compounds isolated from tropical plants of the genus *Calophyllum* occupy a special place and include calanolide, inophyllum, and cordatolide [73,76]. Compounds **50**–**60** and their anti-HIV activity are discussed in detail in reviews [12,73,76,77,78]. The main results are depicted in Table 4. Note that calanolide A **50**, inophyllum B **54,** and inophyllum P **58** were identified as non-nucleoside inhibitors of HIV-1 RT with IC_50_ in the nanomolar range. In addition, calanolide A **50** demonstrated activity against a wide range of HIV strains, including azidotmidine- and pyridinone-resistant HIV strains [73,79,80,81,82]. Currently, calanolide A passed the first phase of clinical trials [12,83,84].

In their review [81], Yu and colleagues considered the structure–activity relationships (SARs) of natural compounds isolated from plants of the genus *Calophyllum*:(1)the presence of bulky substituents at the C(4) position;(2)positions C(10), C(11) and C(12) of the chromanol ring are sensitive to modification;(3)position C(12) must contain a hydrogen bond acceptor.

Subsequently, a systematic study of the SARs of calanolide A analogs led to more active derivatives than parent calanolide A [85,86]. Thus, it was demonstrated that 11-demethyl-12-oxo calanolide A **61** (Figure 10), possessing only one chiral center, has similar inhibitory activity against HIV-1 and a better therapeutic index (EC_50_ = 0.11 μM and TI = 21–169) [87]. 10-Bromomethyl-11-demethyl-12-oxo-calanolide A **62 [85]** and 10-chloromethyl-11-demethyl-12-oxo-calanolide A **63** [86] (Figure 10) showed greater anti-HIV-1 activity at the nanomolar level (EC_50_ = 2.85 nM and TI > 10526 and EC_50_ = 7.4 nM and TI = 1417, respectively). Compounds **61**–**63** demonstrate no detectable toxicity [85]. In addition, compound **63** demonstrated a druggable profile with 32.7% oral bioavailability in rats and extremely high potency against the wild-type HIV-1 and Y181C single mutation of HIV-1 [86].

Pyranocoumarin GUT-70 **64** (Figure 11) was isolated from *Calophyllum brasiliense* stem bark [88,89]. It stabilized plasma membrane fluidity, inhibited HIV-1 entry, and down-regulated the expression of chemokine receptors CXCR4, CD4, and CCR5. GUT-70 also had an inhibitory effect on viral replication through the inhibition of NF-κB [90]. Therefore, this compound may be a dual-functional and viral mutation-resistant reagent. Another important feature of this compound is that GUT-70 inhibited HIV-1 replication in both acutely and chronically infected cells [91]. The antiviral activity (determined in several cell lines) and toxicity (MTT assay) of GUT-70 are presented in Table 5. Unfortunately, despite several mechanisms of action, this compound is characterized by low therapeutic indices.

The seseline scaffold is presented by seselin and suksdorfin-type coumarin, its oxygenated counterpart.

The parent compound, seselin **65** shows moderate anti-HIV activity with EC_50_ = 25.5 μM and CC_50_ = 329.0 μM (TI = 12.9) [22].

Lee and colleagues [72] isolated suksdorfin **66** from the fruit *Lomatium Suksdorfii,* which demonstrated inhibition of HIV-1 replication in H9 cells with EC_50_ = 1.3 μM and TI = 79. Subsequently, a number of suksdorfin 3’,4’-analogs were synthesized, among which 3’,4’-di-O-camphanoyl-*cis*-khellactone (DCK) **67** inhibits HIV-1 replication in H9 cells with EC_50_ = 0.41 nM and TI > 78,000 [74]. Subsequently, the SAR DCK **67** was investigated, which was considered in reviews [12,81]. Table 6 shows anti-HIV active compounds **66**–**67** (Figure 12) in H9 cells.

Among the modified compounds 67 and 68, 3-cyanomethyl-DCK 68e showed weak toxicity (IC_50_ > 37 μM) against HIV-1_IIIB_ in H9 cells. In addition, 68e has anti-HIV activity against wild-type and drug-resistant viral infections in CD4^+^ T cell lines as well as in peripheral blood mononuclear cells. Preclinical trials have shown that compound 68e has moderate oral bioavailability in rats and moderate cell permeability [93].

Thus, among linear and angular pyranocoumarins, the latter are the most promising in terms of developing anti-HIV agents. These compounds are characterized by good antiviral activity, but they have some solubility problems, which seem to be due to the presence of a condensed ring system.

Prenyl derivatives of simple coumarins, pyranocoumarins and furanocoumarins, are genetically related compounds since the biosynthesis of pyranocoumarins and furanocoumarins occurs through cyclization of the isoprene fragment with the aromatic cycle [95,96]. Furthermore, the mode of action of anti-HIV coumarins is often carried out through identical targets, such as RT inhibition.

The unique structure of coumarins determines their low toxicity compared to other heterocycles and, in combination with effective protocols for coumarin synthesis, provides good potential for developing anti-HIV drugs based on the coumarin scaffold.

## 5. Conclusions

Among natural coumarins, there are antiviral compounds with activity against viral enzymes, reverse transcriptase, and integrase. While several types of coumarins (simple coumarins, pyranocoumarins, and furanocoumarins) are active against reverse transcriptase, the only known coumarin with anti-integrase activity is wedelolactone. In addition, simple and angular coumarins are capable of inhibiting cellular factors that regulate HIV-1 replication. Finally, simple coumarins can block the transmission of viral particles from infected macrophages to healthy cells.

Among the anti-HIV coumarin derivatives, the pyranocoumarins—represented by the calanolide and suksdorfin families, as well as simple coumarin C3,C6-diterpenoids—are of the greatest interest. These compounds are characterized by effective inhibition of immunodeficiency virus replication in the nanomolar range, and, in addition, some of them act by several mechanisms. A slight tuning of the structure of the natural compounds allows for obtaining even more active compounds with a therapeutic index of up to 10^8^ for the suksdorfin derivatives.

We reviewed the information on the anti-HIV properties of simple coumarins which allows for the consideration of the coumarin cycle as a privileged framework for the search and further development of anti-HIV compounds. Nevertheless, the unique properties are not limited to anti-HIV activities, but they can also target other RNA-based viruses, among which SARS-CoV-2 currently attracts the most significant attention [97].

## Figures and Tables

**Figure 1 ijms-24-02839-f001:**
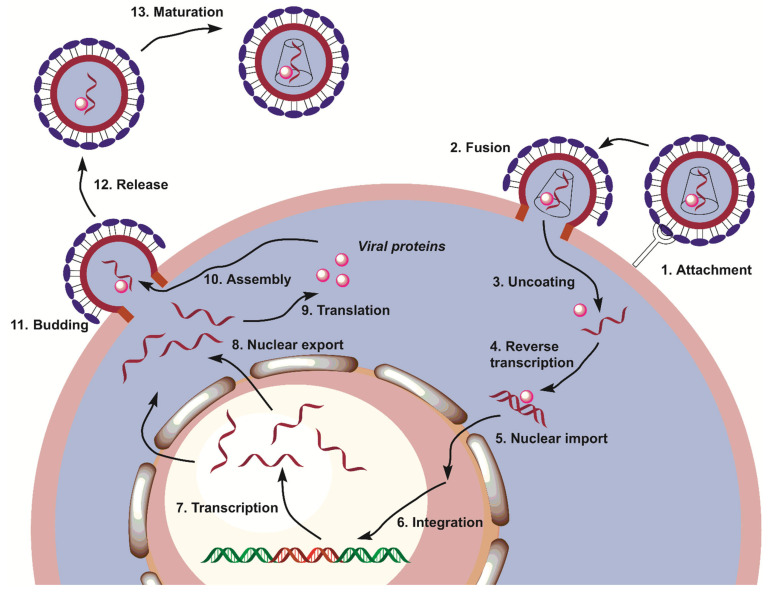
Steps of the HIV-1 replication cycle.

**Figure 2 ijms-24-02839-f002:**
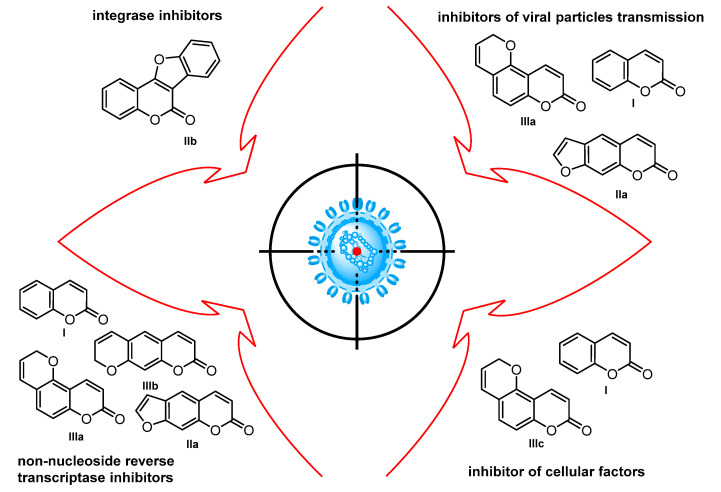
Molecular targets for naturally occurring anti-HIV coumarins.

**Figure 3 ijms-24-02839-f003:**
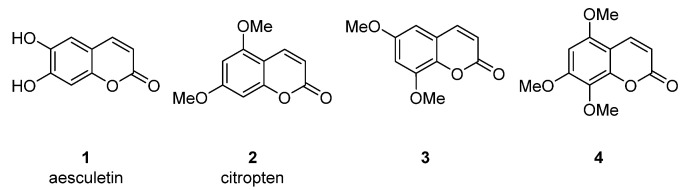
Anti-HIV hydroxy- and methoxycoumarins **1**–**3**.

**Figure 4 ijms-24-02839-f004:**
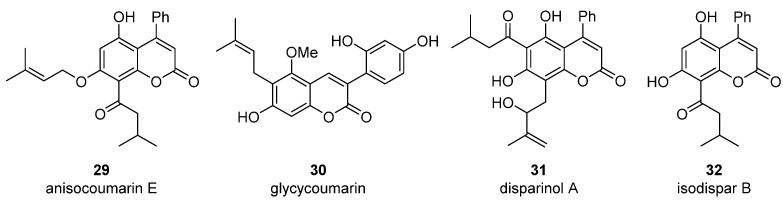
C,O-diterpenoid and phenyl-substituted mono- and diterpenoid coumarins with anti-HIV activity.

**Figure 5 ijms-24-02839-f005:**
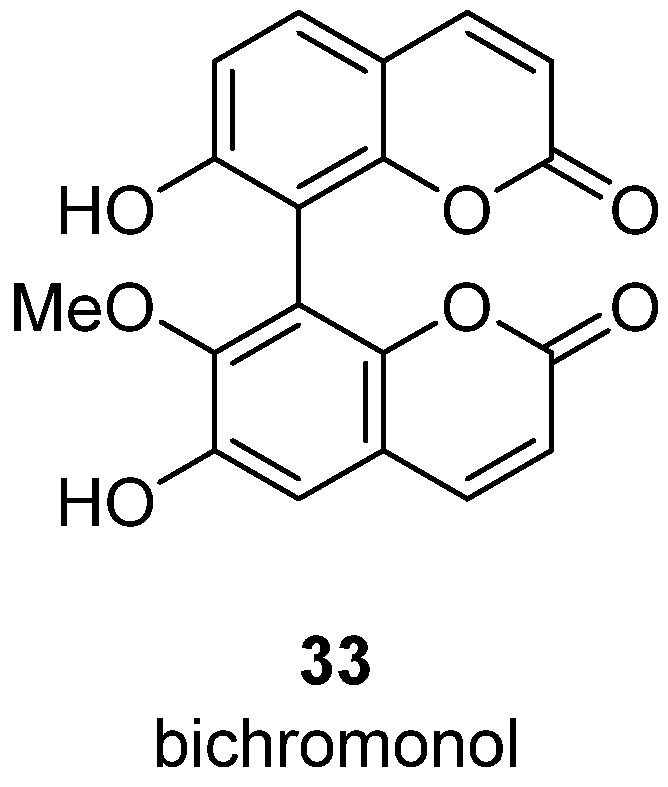
Structure of bichromonol **33**, a coumarin dimer.

**Figure 6 ijms-24-02839-f006:**
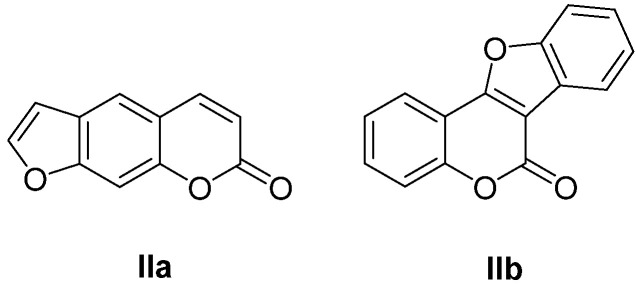
Psoralene (**IIa**)- and coumestan (**IIb**)-type natural furocoumarins.

**Figure 7 ijms-24-02839-f007:**
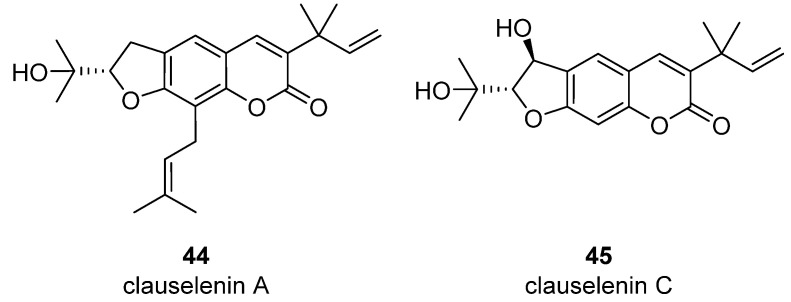
Structure of clauselenins A and C **44** and **45**.

**Figure 8 ijms-24-02839-f008:**
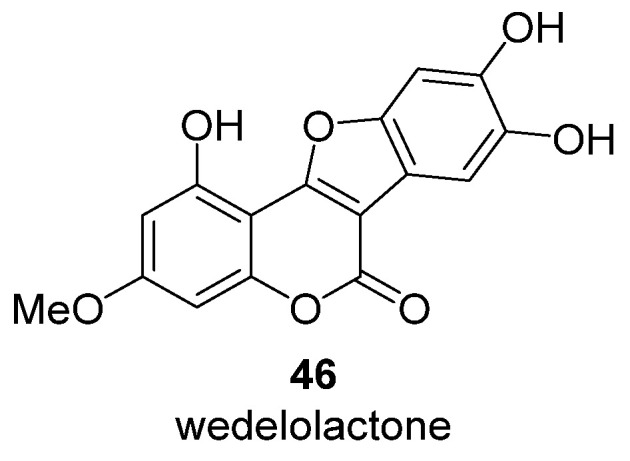
Structure of wedelolactone, a coumestan derivative.

**Figure 9 ijms-24-02839-f009:**
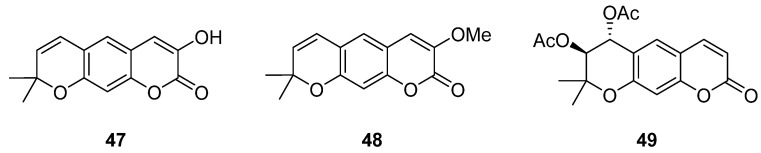
Linear pyranocoumarins with anti-HIV activity.

**Figure 10 ijms-24-02839-f010:**
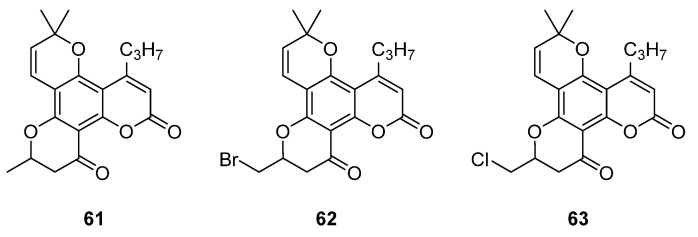
Synthetic calanolide A analogs with anti-HIV activity.

**Figure 11 ijms-24-02839-f011:**
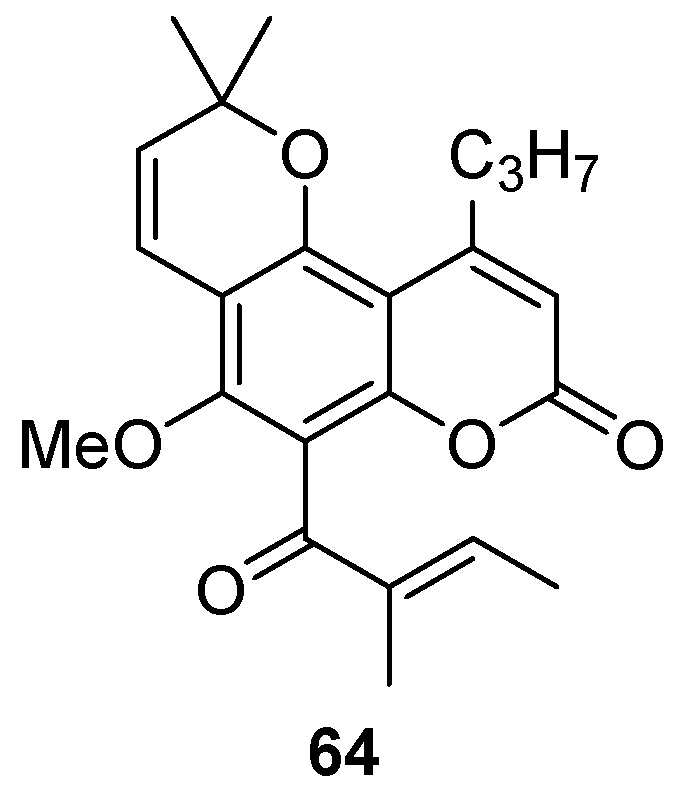
Structure of the anti-HIV pyranocoumarin GUT-70.

**Figure 12 ijms-24-02839-f012:**
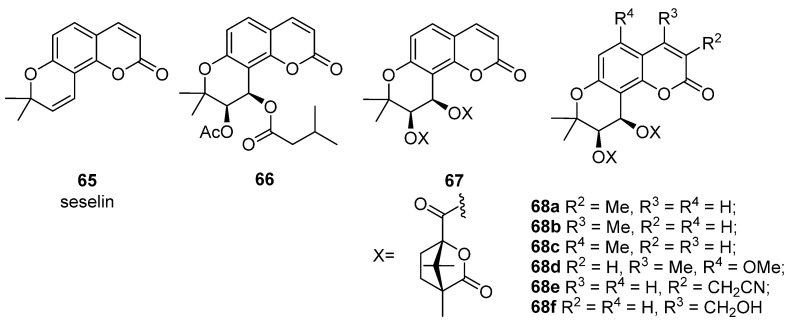
Structures of seselin 65, suksdorfin 66, DCK 67, and its analogues 68.

**Table 3 ijms-24-02839-t003:** Linear furocoumarins with anti-HIV activity.

Compound	Plant Source	Anti-HIV Activity and Toxicity
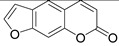 Psoralen **34**	*Prangos tschimganica* [58], *Ruta graveolens* [60], *Dorstenia foetida* [61]	EC_50_ = 0.54 μM CC_50_ = 103 μM TI = 191 [58]
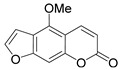 Bergapten **35**	*Zanthoxylum ailanthoides* [23], *Ferula sumbul* [30], *Prangos tschimganica* [58], *gen. Angelica* [62,63]	EC_50_ = 1.63 μM CC_50_ = 115 μM TI = 69.9 [58]
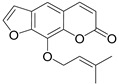 Imperatorin **36**	*Ferula sumbul* [30], *Prangos tschimganica* [58], *gen. Angelica* [62], *,Clausena lansium* [64],	EC_50_ <0.37 μM CC_50_ > 370 μM TI > 1000 [30]
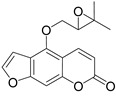 Oxypeucedanin **37**	*Ferula sumbul* [30], *Prangos tschimganica* [58], *gen. Angelica* [62]	EC_50_ = 3.67 μM CC_50_ = 81.8 μM TI = 22.2 [30]
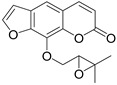 Heraclenin **38**	*Ferula sumbul* [30], *Prangos tschimganica* [58], *gen. Angelica* [62]	EC_50_ = 8.29 μM CC_50_ > 70.3 μM TI = 8.48 [30]
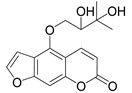 Oxypeucedanin Hydrate **39**	*Ferula sumbul* [30], *Prangos tschimganica* [58], *gen. Angelica* [62]	EC_50_ = 3.29 μM CC_50_ = 6.94 μM TI = 2.11 [30]
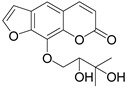 Heraclenol **40**	*Ferula sumbul* [30], *Prangos tschimganica* [58], *gen. Angelica* [62]	EC_50_ = 0.38 μM CC_50_ = 329 μM TI = 870 [30]
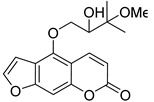 Oxypeucedanin methylate **41**	*Ferula sumbul* [30], *Prangos tschimganica* [58]	EC_50_ = 105 μM CC_50_ = 314 μM TI = 3.00 [30]
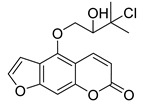 Saxalin **42**	*Prangos tschimganica* [58], *Angelica officinalis* [65]	EC_50_ = 6.99 μM CC_50_ = 81.7 μM TI = 11.7 [58]
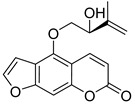 Pabulenol **43**	*Ferula sumbul* [30], *Prangos tschimganica* [58], *gen. Angelica* [62]	EC_50_ = 22.3 μM CC_50_ = 58.4 μM TI = 2.61 [58]

**Table 4 ijms-24-02839-t004:** Compounds **50**–**60** with anti-HIV-1 activity.

Compound	Inhibition of HIV-1 PRP, μM	Anti-HIV Activity in CEM SS Cells and Toxicity
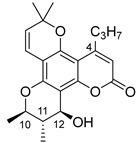 calanolide A **50**	IC_50_ = 0.07 [78]	IC_50_ = 0.1 μM CC_50_ = 20 μM TI = 200 [73,79]
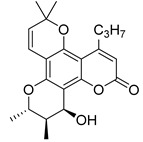 calanolide B **51**		IC_50_ = 0.4 μM CC_50_ = 15 μM TI = 37 [73,76,79]
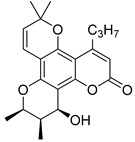 calanolide C **52**		no significant activity CC_50_ = 30 μM [79]
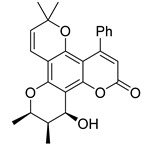 inophyllum A **53**		IC_50_ = 30 μM [80]
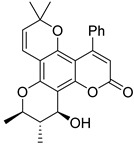 inophyllum B **54**	IC_50_ = 0.038 [73]	IC_50_ = 1.4 μM CC_50_ = 55 μM TI = 39 [73,80]
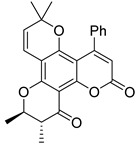 inophyllum C **55**		low activity CC_50_ = 18 μM [80]
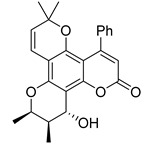 inophyllum D **56**		low activity CC_50_ = 15 μM [80]
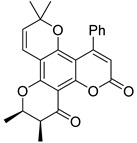 inophyllum E **57**		low activity CC_50_ = 6.2 μM [80]
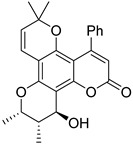 inophyllum P **58**	IC_50_ = 0.130 [73]	IC_50_ = 1.6 μM CC_50_ = 55 μM TI = 16 [80,81]
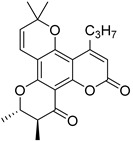 (+)-oxocalanolide **59**		IC_50_ HIV-1 (III_B_) = 1.0 μM, CC_50_ HIV-1 (III_B_) > 10 μM IC_50_ HIV-1 (R_F_) = 0.9 μM, CC_50_ HIV-1 (R_F_) > 10 μM IC_50_ HIV-1 (SK1) = 0.17 μM, CC_50_ HIV-1 (SK1) > 10 μM IC_50_ HIV-2 (ROD) = 15.9 μM, CC_50_ HIV-1 (ROD) > 13.8 μM [81]
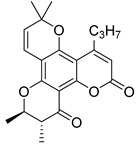 (–)-oxocalanolide **60**		IC_50_ HIV-1 (III_B_)= 1.88 μM, CC_50_ HIV-1 (III_B_) > 10 μM IC_50_ HIV-1 (R_F_) = 3.41 μM, CC_50_ HIV-1 (R_F_) > 10 μM IC_50_ HIV-1 (SK1) = 0.27 μM, CC_50_ (SK1) = 55 μM [81]

**Table 5 ijms-24-02839-t005:** Antiviral activity and toxicity of GUT-70.

Cell Line	CC_50_, μM	EC_50_, μM	TI (CC_50_/EC_50_)
U1 (PMA)	8.44	3.48	2.43
U1 (TNF-α)	8.44	4.32	1.95
Molt-4	>10	3.41	>2.93
TZM-bl	>10	–	–

Reprinted/adapted with permission from Ref. [91]. 2013, Elsevier.

**Table 6 ijms-24-02839-t006:** Anti-HIV activity of suksdorfin and DCK analogs in H9 cells.

Compound	Anti-HIV Activity, EC_50_ μM (TI)	Compound	Anti-HIV Activity, EC_50_ μM (TI)
Seselin 65	25.5 (12.9) [22]	68c	2.4 × 10^–7^ (4 × 10^8^) [92]
Suksdorfin 66	1.3 (79) [72]	68d	7.2 × 10^–6^ (2.1 × 10^7^) [93]
DCK 67	2.5 × 10^–4^ (1.4 × 10^5^) [92]	68e	2.4 × 10^–3^ (1.5 × 10^4^) [94]
68a	5.3 × 10^–5^ (2 × 10^6^) [92]	68f	4 × 10^–3^ (6 × 10^3^) [94]
68b	1.8 × 10^–6^ (6.9 × 10^7^) [92]		

## Data Availability

Not applicable.

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
