# Peer review of "Plant Coumarins with Anti-HIV Activity: Isolation and Mechanisms of Action"

_ijms, 2023, doi:10.3390/ijms24032839_

Round 1
Reviewer 1 Report
Major points
1. Authors described anti-HIV activities of coumarin compounds and their derivatives in the text and list. They inhibit HIV protease, integrase, reverse transcriptase, viral DNA replication, vpr, sp1-related genes (cell cycle arrest), Tat, Rev, and glycosylation. However, the underlying mechanism of some of them is not clearly described. Authors are better to describe it more details.
2. Authors need to describe the cytotoxicity of coumarin compounds and their derivatives if published papers are available.
Minor points
1. Lines 66-67, 71, and Tables: F. ornus, M. esculenta 66, A. capillaries and Z. ailanthoides: Authors have to describe the full name of plant species (binomial nomenclature) when their scientific names were used for the first time. And then, authors can use the initials of the genus name. These mistakes were frequently found throughout the text. Please, check the note that I mentioned.
2. There is no explanation of compound 3 in text and Fig. 2.
3. Line 148: ‘compound 13’ ? Is it compound 31?
4. Line 157: ‘demonstrane’ to demonstrated
5. In the text, some plant names are correctly written, but others are written by normal letters. They should be changed to italic letters.
6. Line 247: change ‘Calophyllum Brasiliense’ to Calophyllum brasiliense
7. Line 247: add stem bark before or after Calophyllum Brasiliense
8. Line 257: change ‘T.T.-Y. Lee and colleagues’ to Lee and colleagues
Reviewer 2 Report
The manuscript entitled "Plant coumarins with anti-HIV activity: isolation and mechanisms of action" in which the authors focused and summarized the literature data on anti-HIV activity of plant coumarins with emphasis on isolation and the mechanism of their antiviral actions showing several kinds and mechanisms.
The work is understandable and the topic is important. The scientific narrative is well structured and flows naturally from one idea to the next.
However, this paper suffers from few shortcomings that if modified would make the manuscript very suitable for publication in International Journal of Molecular Sciences.
Shortcomings:
1- The authors write “The unique structure of coumarins determines low toxicity compared to other heterocycles and, in combination with effective protocols for coumarin synthesis [87,88]”. They mentioned the low toxicity of some coumarins in some studies. However, they didn’t talk about if there are adverse effects of coumarins in experimental studies. Please talk briefly about some adverse effects of coumarins.
2- Are there any clinical trials about the therapeutic use of coumarins in HIV patients? Please add briefly if there are.
3- The authors write “These compounds are characterized by good antiviral activity, but they have some solubility problems, which seem to be due to the presence of a condensed ring system”. Please add how do the researchers overcome these solubility problems from literature?
4- Below is some advice to change (related to typos and language):
· “EtOH extract in line 114. Please write the abbreviation.
· “3-(1,1-Dimethylallyl)-8-hydroxy-7-methoxycoumarin 8 (Table 2), the only representative of C3-monoterpenoids, inhibit HIV viral replication”. Please correct “inhibits”.

Round 2
Reviewer 1 Report
The revised of the manuscript is very improved, and therefore it is ready to go publication in IJMS.